# Look-alike modelling in violence-related research: A missing data approach

**Estela Capelas Barbosa** [1] *, **Niels Blom** [2], **Annie Bunce** [3]

**1** Population Health Sciences, Bristol Medical School, University of Bristol, Bristol, United Kingdom,
**2** University of Manchester, Manchester, United Kingdom, **3** Violence and Society Centre, City St George's,
University of London, London, United Kingdom

* e.capelasbarbosa@bristol.ac.uk

Look-alike modelling in violence-related research:
A missing data approach. PLoS ONE 20(1):
e0301155. https://doi.org/10.1371/journal.
pone.0301155

University, NIGERIA

**Data Availability Statement:** Data for the Crime
Survey for England and Wales can be downloaded
by registered researchers from UK Data Service
(https://ukdataservice.ac.uk/find-data/). The data
from RCEW are not publicly available due to legal

## Abstract

Violence has been analysed in silo due to difficulties in accessing data and concerns for the
safety of those exposed. While there is some literature on violence and its associations
using individual datasets, analyses using combined sources of data are very limited. Ideally
data from the same individuals would enable linkage and a longitudinal understanding of
experiences of violence and their (health) impacts and consequences. This paper aims to
provide proof of concept to create a synthetic dataset by combining data from the Crime Sur-
vey for England and Wales (CSEW) and administrative data from Rape Crisis England and
Wales (RCEW), pertaining to victim-survivors of sexual violence in adulthood. Intuitively, the
idea was to impute missing information from one dataset by borrowing the distribution from
the other. In our analyses, we borrowed information from CSEW to impute missing data in
the RCEW administrative dataset, creating a combined synthetic RCEW-CSEW dataset.
Using look-alike modelling principles, we provide an innovative and cost-effective approach
to exploring patterns and associations in violence-related research in a multi-sectorial set-
ting. Methodologically, we approached data integration as a missing data problem to create
a synthetic combined dataset. Multiple imputation with chained equations were employed to
collate/impute data from the two different sources. To test whether this procedure was effec-
tive, we compared regressions analyses for the individual and combined synthetic datasets
on binary, continuous and categorical variables. We extended our testing to an outcome
measure and, finally, applied the technique to a variable fully missing in one data source.
Our results show that the effect sizes for the combined dataset reflect those from the dataset
used for imputation. The variance is higher, resulting in fewer statistically significant esti-
mates. Our approach reinforces the possibility of combining administrative with survey data-
sets using look-alike methods to overcome existing barriers to data linkage.

## Introduction

It has been established for over 20 years that violence is a complex social problem and a public
health issue [1–3], with implications for the health and social care systems, police and justice
systems [4], as well as significant productivity losses for those who experience it [5, 6].

restrictions agreed in the data sharing process with Rape Crisis England and Wales. These legal and data protection obligations are outlined in DPIA provided with this submission due to concerns for the safety and risk of further- and re-victimisation of victims of sexual violence who sought support from RCEW. The data that support the findings of this study can be made available on reasonable request from the corresponding author, ECB, if consented by Rape Crisis England and Wales. The data protection officer at City, University of London is Emma White and she can respond to queries around data protection at dataprotection@city.ac.uk.

**Funding:** This paper is a result of VISION research, which is supported by the UK Prevention Research Partnership (Violence, Health and Society; MR-VO49879/1). VISION is a Consortium funded by the British Heart Foundation, Chief Scientist Office of the Scottish Government Health and Social Care Directorates, Engineering and Physical Sciences Research Council, Economic and Social Research Council, Health and Social Care Research and Development Division (Welsh Government), Medical Research Council, National Institute for Health and Care Research, Natural Environment Research Council, Public Health Agency (Northern Ireland), The Health Foundation, and Wellcome. The views expressed are those of the researchers and not necessarily those of the UK Prevention Research Partnership or any other funder. 'The funders had no role in study design, data collection and analysis, decision to publish, or preparation of the manuscript.

**Competing interests:** All authors declare no conflicts of interest.

Analysing data collected by these systems can aid understanding of the problem of violence and how to respond to it. In social research, analysing administrative records together with survey data has already enabled better measurements of violence experiences, capturing experiences of both victim-survivors and perpetrators across multiple points in time and social and economic domains [7].

Although some violence-related research has been carried out using matched or combined emergency departments and police data [8–11], most studies in violence-related research analyse data in silo due to difficulties in accessing data and concerns for the safety of those exposed [12, 13]. Particularly, data from third sector voluntary specialist support services for victims or perpetrators of violence has, to our knowledge, not been linked or combined with other datasets, as these services are keen to provide person-centred trauma-informed care and fear that information on their service users may be used against them in courts or by immigration authorities [14, 15].

From an analytical viewpoint, ideally, data from the same individuals would enable linkage and a longitudinal understanding of experiences of violence and their (health and inequalities) impacts. However, given safety concerns, data on people who have experienced violence is often pseudonymised before being made available for researchers, meaning records across sectors pertaining to the same individuals cannot be linked. Look-alike profiling may provide an innovative and cost-effective approach to exploring patterns and associations in violence-related research in a multi-sectorial setting.

Look-alike modelling has been extensively used to identify similar and new costumer and consumer target groups in marketing, e-commerce and advertising [16–18]. We apply costumer look-alike principles to violence-related research. Our goal is to propose an innovative method for data integration in this particularly sensitive research area, to move beyond silo analyses, which could also be used in other research areas with similar issues. Effectively, this method allows for integrating additional information into one dataset based on its distribution and associations in another dataset, creating a new (synthetic) dataset. This methodology could also be used in other fields of social and economic research, where issues regarding pseudonymisation and missing information are also present.

In this paper, we approached the problem of data integration and look-alike profiling as a missing data problem, although we acknowledge that several other approaches are possible. We combined data from the Crime Survey for England and Wales (CSEW) with administrative data from three Rape Crisis Centres (RCC) in England, which are part of a Rape Crisis England and Wales (RCEW), focussing on victim-survivors of sexual violence in adulthood, in line with the understanding that a benefit of linking administrative and survey data is the improvement in imputation methods to fill in missing values in surveys [19]. Multiple imputation with chained equations were employed to collate and integrate data from these two different sources producing a synthetic dataset.

## Theoretical framework

In theory, look-alike modelling is based on the principle that similar individuals have similar behaviours. While in economics this normally refers to consumption behaviour, for people experiencing violence it refers to their trajectories and help-seeking behaviours. Therefore, to explore similarities between individuals, one needs to look at socio-economic and demographic variables, as well as violence experience. Mathematically, in two different datasets A and B, there are $a_{ij}$ and $b_{ij}$ individual records. These records can be compared in multiple variables $k$ to ascertain how similar their look-alike profiles are. Each component-wise or variable-wise comparison relies on a vector $C_{i,j} = [c_1{}^{i,j}, c_2{}^{i,j}, \ldots, c_k{}^{i,j}]$ that effectively produces a

*comparison function* looking at the values of the record component or variable $k$ in the two records $a_{ij}$ and $b_{ij}$. In order to approach this data integration problem as a missing data problem, one relies on a sequence of univariate imputation models, with fully conditional specifications of prediction equations. Formally, for imputation variables $X_1, X_2, \ldots, X_p$ and complete independent predictors C, so that:

$$
\begin{aligned}
X_1^{(t+1)} &\sim g_1(X_1 | X_2^{(t)}, \ldots, X_p^{(t)}, C, \varphi_1) \\
X_2^{(t+1)} &\sim g_2(X_2 | X_1^{(t+1)}, X_3^{(t)}, \ldots, X_p^{(t)}, C, \varphi_2) \\
&\cdots \\
X_p^{(t+1)} &\sim g_p(X_p | X_1^{(t+1)}, X_2^{(t+1)}, \ldots, X_{p-1}^{(t+1)}, C, \varphi_p)
\end{aligned}
\tag{1}
$$

Where t are iterations that converge at t = T and $\varphi_j$ are the corresponding model parameters prior [20]. In our study, we created the vector $C_{i,j}$ based on the following variables: type of sexual violence experienced (type of SV), relationship to the perpetrator, health impact, employment status, housing tenure, number of dependants, relationship status (usually referred to as marital status in social research), ethnicity, age and gender. These variables were selected as they are considered to influence the journey of victim-survivors of sexual violence and their help-seeking behaviour.

Traditionally, multiple imputation (MI) is used to address missingness of data by generating plausible values derived from distributions and relationships among observed variables [21]. While MI has been widely used in statistical and economic analysis of clinical trials [22] and more recently social research [23], to our knowledge, it has not been used to produce a synthetic dataset. Our multiple imputation approach to data integration recognises that the reason for missing data may be different for each dataset A and B. This is particularly true in our empirical application, since we are using a population-level survey (CSEW) and administrative records from a victim support service (RCEW). Furthermore, while datasets A and B are completely independent in our case, the reasons for missingness may be correlated, as disclosing sexual abuse is still stigmatised in society [24–26]. Finally, our approach recognises that the variables or covariates used for imputation may have non-normal distributions [27, 28].

Procedurally, multiple imputation replaces each missing value with a set of plausible values. Following Bayesian rules, the imputed values are drawn based on the conditional distribution of the missing observations given the observed data, reflecting the uncertainty associated with the missing data itself and parameters estimated in the imputation model [29]. Mathematically, let $f_{ij}$ represent the variable you are interested in imputing for the $i$th individual within the $j$th cluster. In this case, $C_{i,j} = [c_1^{i,j}, c_2^{i,j}, \ldots, c_k^{i,j}]$, the *comparison function* and $D_j$, the cluster-level vector of covariates, are the predictors of missingness in variable $f$ at individual and cluster-levels respectively. Then, a MI model can be specified as:

$$
f_{ij} = \beta^f C_{ij} + \gamma^f D_j + \varepsilon_{ij}^f
\tag{2}
$$

Where $\beta$ and $\gamma$ are the vectors of the regression coefficients corresponding to individual and cluster-level covariates. The model assumes that the error term ($\varepsilon$) is normally distributed with variance $\sigma^2$. The imputation procedure generates multiple values for each missing observation based on the distributions for $\beta$, $\gamma$ and $\sigma$ conditioned on observed data. By combining two datasets A and B, based on the vector $C$ and using multiple imputation, we are applying a look-alike modelling approach that may enable imputation of partially and completely missing data into a complete combined synthetic dataset.

## Methods

We aimed to test our proposed approach to data integration by combining survey data from the Crime Survey for England and Wales (CSEW) with administrative data from Rape Crisis England & Wales (RCEW), focussing on victim-survivors of sexual violence in adulthood. Intuitively, the idea was to impute missing information from one dataset by borrowing the distribution from the other. In our analyses, we borrowed information from CSEW to impute missing data in the RCEW administrative dataset, creating a combined synthetic RCEW-CSEW dataset.

This research was reviewed and approved by the IMJEE (International Politics, Music, Journalism, Economics, and English) research ethics committee from City, University of London (ETH2122-2023 and ETH2122-0299). Informed verbal consent regarding future use of their data for research was obtained by case workers from Rape Crisis centres while working with service users and recorded in their case management system, in line with their a non-intrusive approach to data collection whereby only what is appropriate is asked and/or what survivors choose to disclose is recorded [30]. Both the CSEW and RCEW datasets were accessed by the authors between October 2022 and November 2023 for the purpose of this study.

### Datasets

The CSEW, previously known as the British Crime Survey, is a nationally representative face-to-face victimisation survey of about 35 thousand to 46 thousand respondents per survey wave, which started biannually from 1982 before becoming an annual survey from 2001 [31]. The CSEW asks people aged 16 and over about their experience with household and personal crimes in the twelve months prior to the interview. Considering our focus on sexual violence, we only included individual level data from respondents who had reported being a victim-survivor of rape, attempted rape, wounding with sexual motive, and indecent assault. In order to include a sufficient number of incidents of sexual violence to do the data integration, we used CSEW data from 2001 to 2020.

The RCEW data comes from three RCCs in a region in eastern England and is based on routinely collected administrative data recorded in a centralised case management data system between April 2016 and March 2020. Information is self-reported by victim-survivors upon initial contact with RCEW, most commonly over the phone but sometimes online or face-to-face, and data are inputted to the RCEW database by frontline support workers. Rape Crisis centres collect individual level data for their service users in pre-determined coding categories based on a person-centred non-intrusive principle, which means frontline workers only ask questions that are appropriate, or rely on information victim-survivors choose to disclose [30]. Information collected typically includes socio-demographic and protected characteristics (gender, age, disability, ethnicity, nationality, sexuality, religion, marital status, accommodation, employment, language, immigration status, socioeconomic status), experiences of sexual violence and abuse (SVA), victim-perpetrator relationship, impacts from experience of SVA, risk level, referral routes, engagement with different (statutory and non-statutory) services and contact with the criminal justice system. Data on experiences of SVA are collected in two main ways; information is gathered on the 'presenting incident' (the main experience of violence the victim-survivor is seeking support for at the time of initial contact with RCEW), and elsewhere in the database further details can be entered under 'incident summary' about separate 'incidents' or experiences of violence, if disclosed [30]. Most information is inputted into their case management system at the point of intake based on the victim-survivor's report and, where necessary, the assessment of the support worker. However, further information on the abuse can be collected and recorded at any point during the support journey, as appropriate. Case

management and criminal justice data are collected in separate parts of the system, however, data are recorded under a client identification number, making it possible to merge case management and criminal justice data.

Considering our focus on sexual violence, we selected respondents (CSEW) or service users (RCEW) who have reported being a victim-survivor of rape (including attempted) or another form sexual violence and abuse (which included indecent assault and wounding with sexual motive). We selected respondents/service users with no missing values on vector *C* variables, which led to a sample of 1,232 incidents from 1,111 individuals in the CSEW, and 6,102 referral cases from 5,333 individuals in RCEW. In RCEW data, it included data for individuals who accessed the service more than once. These two datasets (CSEW and RCEW) are sufficient to achieve our aim of creating a combined synthetic dataset and no statistical constraints were relaxed while conducting our empirical application.

## The comparison vector (C) variables

As previously mentioned, we created the vector $C_{i,j}$ based on the variables that were considered to influence victim-survivors' journeys and help-seeking behaviour the most. Thus, we needed to harmonise the following variables across CSEW and RCEW data: type of sexual violence experienced, relationship to the perpetrator, health impact, employment status, housing tenure, number of dependants, relationship status, ethnicity, age and gender.

The type of sexual violence experienced in the CSEW was categorised into crime codes by professional coders based on respondents' responses to survey questions and narrative description of the incident. The categories are aimed to align with Home Office categorisation. We selected the following reported offences: rape, serious wounding with sexual motive, other wounding with sexual motive, attempted rape, and indecent assault. We categorised these into rape (including attempted) or some other form of sexual violence. In the RCEW data, sexual violence was categorised based on the information recorded at intake under 'presenting incident' and 'incident summary'. Once again, we categorised these into broader categories: rape (as an adult, including attempted rape); and some other form of sexual violence (including sexual assault, assault by penetration, voyeurism, sexual bullying, penetration by object, gang related sexual violence, forced sexual activity in public, exposed to sexual images, sexual harassment and sexual exploitation). Victim-survivors accessing RCEW services for other types of violence or abuse were excluded, including rape or sexual abuse during childhood.

For the variable victim-perpetrator relationship, respondents to CSEW were first asked whether they knew the perpetrator, and if so, what their relationship was at the time of the incident. The RCEW data recorded who the primary perpetrator was. This was categorised into domestic (such as [former] intimate partner or family member), acquaintances (including friends, colleagues), and strangers. If multiple perpetrators were mentioned, it was coded as the closer relationship (e.g. prioritising domestic over acquaintances).

The health impact of the incident was assessed in the CSEW by whether they were bruised, scratched, cut or injured in any way as a result of the incident. The health impact was measured in the RCEW data using information recorded under 'incident impact' and 'impact summary', for which we included physical health impacts of memory loss, physical injuries and body problems, gynaecological disorder, and sexually transmitted infection. While these do not match directly between the two datasets, we only included a binary in our empirical application for whether there was (yes/no) a health impact on the victim-survivor.

Relationship status was categorised into whether respondents were in a co-residential relationship (either married or cohabiting), single/non-resident partner/widowed, or separated or divorced in the CSEW and RCEW. Ethnicity was coded as White and non-White, as further differentiation led to too small numbers in some categories. However, we acknowledge that the ethnicity categorisation of White/non-White may be problematic and any conclusion in this respect, limited [32]. Employment status in both datasets was assessed by whether people were employed, unemployed, students, or outside the labour force (e.g. a homemaker, retired, or unable to work due to illness). Gender was asked as whether the respondent was male or female in the CSEW. We acknowledge that female / male are correct categories for sex not necessarily gender. But we used the categories as asked by the CSEW as proxies for women / men. In RCEW data, more detailed responses are given, including transgender female and transgender male, which were recoded into men and women. Finally, age was measured numerically in both datasets and we included in our analyses people over the age of 16. S1 Table in the supporting information summarises how variables were harmonised.

## Analytical strategy

To test whether approaching look-alike modelling as a missing data problem was effective, we compared regression analyses for the two datasets (CSEW and RCEW) and the combined synthetic dataset, which imputed data based on the comparison vector. As a proof of concept, we tested the approach using variables of different types (binary and continuous) that are observed in both datasets. Formally, our approach had three steps. First, we specified the same linear (OLS) or logistic regression (as appropriate) for dataset A (RCEW) and dataset B (CSEW). We then assumed one variable was missing from the combined integrated synthetic dataset by generating a completely missing variable for dataset A, which we imputed, using multiple imputation with chained equations (MICE), based on the observed values for the combination vector in both datasets. This effectively imputed the (assumed) missing variable in dataset A based on the distribution and associations with other variables of the combination vector in dataset B.

We carried out this exercise for four variables, two that are very similarly measured–age (continuous) and gender (binary), one that is differently measured across datasets–health impact (binary), and lastly, we illustrated the potential of this method of combining data in a real-life application to a variable that only appears in one dataset (CSEW)–frequency of abuse (count). We acknowledge that the first two tests, using age and gender, are not particularly interesting from an analytical standpoint. Nonetheless, we wanted to start off with variables that were objectively measured as much as possible.

## Results

### Profiles comparison of sexual violence victim-survivors in CSEW and RCEW

Before conducting our look-alike exercises, we compared the profiles of sexual violence victim-survivors in CSEW and RCEW datasets (Table 1). The table shows some meaningful differences between the individuals pertaining to each dataset. Particularly, only 32% of sexual violence victim-survivors in the CSEW had been victims of rape, compared to 71% in the RCEW data. Relationship to the perpetrator was more likely to be domestic in the RCEW data compared to the CSEW (48% *vs* 25%, respectively) and perpetrators were far more likely to be strangers or to be unknown in the CSEW (42%), compared to only 12% of records in the RCEW dataset.

**Table 1. Descriptive statistics of sexual violence victim-survivors in the Crime Survey for England and Wales (CSEW) and Rape Crisis England & Wales (RCEW) datasets.**

| | CSEW | | RCEW | |
|---|---|---|---|---|
| | **%** | **Mean (SD)** | **%** | **Mean (SD)** |
| *Type of sexual violence* | | | | |
| Rape | 32.4 | | 70.7 | |
| Other sexual violence and abuse | 67.6 | | 29.3 | |
| *Victim-perpetrator relationship* | | | | |
| Domestic | 24.6 | | 48.2 | |
| Acquaintance | 33.2 | | 40.2 | |
| Stranger or unknown | 42.2 | | 11.6 | |
| *Physical health impact* | | | | |
| No injury | 60.9 | | 94.0 | |
| Injury | 39.1 | | 6.0 | |
| *Gender* | | | | |
| Male | 9.6 | | 7.9 | |
| Female | 90.4 | | 92.1 | |
| *Relationship status* | | | | |
| Married/Cohabiting | 21.7 | | 16.8 | |
| Single/non-resident relationship/Widowed | 57.0 | | 71.7 | |
| Separated/Divorced | 21.4 | | 11.6 | |
| *Ethnicity* | | | | |
| White | 91.6 | | 91.0 | |
| Non-White | 8.4 | | 9.0 | |
| *Employment status* | | | | |
| Employed | 56.4 | | 35.2 | |
| Unemployed | 7.6 | | 38.2 | |
| Outside labour force | 30.4 | | 11.7 | |
| Student | 5.6 | | 14.9 | |
| *Housing tenure* | | | | |
| Homeowner/lives in own home | 34.6 | | 41.8 | |
| Renter | 59.8 | | 18.4 | |
| Other | 5.6 | | 39.7 | |
| *Age* | | 32.9 (12.4) | | 34.1 (12.9) |
| *Nr of dependents* | | 0.6 (1.0) | | 0.8 (1.2) |
| Observations (N) | | 1,232 | | 6,102 |

Source: Crime Survey for England and Wales (2001–2020) and Rape Crisis England & Wales (2016–2020).

N = sample size

Furthermore, the CSEW recorded a physical injury in 39% of incidents, while this appeared in only 6% of cases in the RCEW dataset, which might reflect the different measurements of physical health impact between these two data sources. Finally, there are some differences in relationship status and employment status, with more single or widowed people in RCEW data and more separated or divorced people in CSEW, and more unemployed people and students in RCEW when compared to CSEW.

## Look-alike empirical application

Our first empirical application exercise pretended the variable *age* was missing from the combined dataset. Thus, we stipulated our comparison vector (C) as:

$$C_{i,j}^1 = f[\text{type of SV, perpetrator relationship, health impact, employment status,}$$

$$\text{housing tenure, number of dependants, relationship status, ethnicity and gender}] \quad (3)$$

In this scenario, a possible research question would be: what is the relationship between age (as a dependent variable) and type of sexual violence experienced, relationship to the perpetrator, health impact, employment status, housing tenure, number of dependants, relationship status, ethnicity and gender in the RCEW, in the CSEW and in the combined synthetic RCEW-CSEW datasets? More realistically, such an imputed dataset could be used to answer questions such as how is age related to type of sexual violence victimisation among people accessing specialist support services. Table 2 presents the results of a linear regression (OLS), looking at the associations between age as the dependent variable, and the independent variables for dataset A (RCEW), dataset B (CSEW) and the complete combined dataset inputting age based on our proposed approach. When comparing the associations with age between the original datasets, and the imputed synthetic dataset based on the variation observed in B, the results show that the effect sizes and direction for the imputed data reflects the results from the dataset used as the basis for imputation. For example, the type of SV was not associated with age in the original RCEW, but was in the CSEW. The imputed synthetic dataset reflects the CSEW dataset in that those who were victim-survivors of rape were younger on average. Reversely, while the perpetrator being an acquaintance compared to domestic was associated with younger people in RCEW, this was not the case for the CSEW, where no significant association was found, which was also the case in the imputed synthetic dataset. One coefficient was significantly related to age in both datasets, but not in the imputed version (stranger/ unknown perpetrator). For all independent variables / controls, the standard errors were similar between the CSEW and the imputed synthetic dataset, which additional testing indicates is due to two opposing mechanisms which (partially) cancel each other out. That is, on the one hand, imputation may result in larger standard errors due to the uncertainty around the imputation; on the other hand, the bigger sample size of the imputation sample leads to smaller standard errors.

We then tested the approach on a binary variable, *gender*. For this, we stipulated that the comparison vector was specified as:

$$C_{i,j}^2 = f[\text{type of SV, perpetrator relationship, health impact, employment status,}$$

$$\text{housing tenure, number of dependants relationship status, ethnicity and age}] \quad (4)$$

In this scenario, a possible research question would be: what is the relationship between gender (as a dependent variable) and type of sexual violence experienced, relationship to the perpetrator, health impact, employment status, housing tenure, number of dependants, relationship status, ethnicity and age in the RCEW, in the CSEW and in the combined synthetic RCEW-CSEW datasets? More realistically, such an imputed dataset could be used to answer questions such as how is gender related to type of sexual violence victimisation among people accessing specialist support services. Table 3 shows the results of logistic regressions looking at the associations between gender as a dependent variable and the independent variables for dataset A (RCEW), dataset B (CSEW) and the complete combined synthetic dataset. Similarly

**Table 2. Associations between age and other variables in RCEW data, CSEW data, and the imputed synthetic dataset.** OLS models.

| | Dataset A: RCEW original | Dataset B: CSEW | Synthetic: Dataset A imputed based on Dataset B |
|---|---|---|---|
| | B(SE) | B(SE) | B(SE) |
| *Sexual violence (Ref: Other)* | | | |
| Rape | -0.405 | -1.949** | -1.761** |
| | (0.298) | (0.708) | (0.683) |
| *Victim-perpetrator relationship (Ref: domestic)* | | | |
| Acquaintance | -1.639*** | -0.127 | 0.351 |
| | (0.289) | (0.820) | (0.656) |
| Stranger or unknown | -1.674*** | -1.658* | -0.747 |
| | (0.439) | (0.833) | (0.899) |
| *Gender (Ref: Female)* | | | |
| Male | 3.161*** | 2.831** | 2.241** |
| | (0.501) | (1.009) | (0.686) |
| *Health impact (Ref: No injury)* | | | |
| Injury | 0.634 | 0.531 | 0.918 |
| | (0.559) | (0.676) | (1.262) |
| *Relationship status (Ref: Married/cohabiting* | | | |
| Single/widowed | -8.157*** | -5.234*** | -5.410*** |
| | (0.376) | (0.765) | (0.452) |
| Separated/divorced | 2.289*** | 7.999*** | 7.382*** |
| | (0.512) | (0.916) | (1.002) |
| *Ethnicity (Ref: White)* | | | |
| Not White | -0.697 | 0.382 | 0.758 |
| | (0.463) | (1.054) | (1.361) |
| *Employment status (Ref: Employed)* | | | |
| Unemployed | 0.605 | -2.299* | -2.120** |
| | (0.330) | (1.154) | (0.799) |
| Outside labour force | 9.383*** | 4.033*** | 4.118*** |
| | (0.459) | (0.690) | (0.822) |
| Student | -10.746*** | -6.385*** | -6.500* |
| | (0.427) | (1.335) | (2.811) |
| *Housing tenure (Ref: Homeowner)* | | | |
| Renter | 1.380*** | -4.351*** | -4.768*** |
| | (0.382) | (0.659) | (0.709) |
| Other | 2.651*** | -9.545*** | -9.866*** |
| | (0.324) | (1.363) | (1.582) |
| *Nr of dependent* | -0.625*** | -2.486*** | -2.274*** |
| | (0.124) | (0.318) | (0.681) |
| Constant | 40.074*** | 39.097*** | 38.591*** |
| | (0.456) | (1.059) | (1.368) |
| Observations (N) | 6,102 | 1,232 | 6,102 |

Source: based on CSEW and RC datasets

*** p<0.001

** p<0.01

* p<0.05. N = sample size.

Results presented as regression coefficients (for consistency) followed by standard errors (SE) in brackets.

**Table 3. Associations between gender and other variables in RCEW data, CSEW data, and the imputed synthetic dataset.** Logistic regression models.

| | Dataset A: RCEW original | Dataset B: CSEW | Synthetic: Dataset A imputed dependent based on dataset B |
|---|---|---|---|
| | B(SE) | B(SE) | B(SE) |
| *Sexual violence (Ref: Other)* | | | |
| Rape | -1.113*** | -0.840** | -0.747* |
| | (0.100) | (0.302) | (0.327) |
| *Victim-perpetrator relationship(Ref: domestic)* | | | |
| Acquaintance | 0.646*** | 0.700 | 0.921*** |
| | (0.106) | (0.404) | (0.197) |
| Stranger or unknown | 0.189 | 1.130** | 1.352*** |
| | (0.181) | (0.394) | (0.303) |
| *Health impact (Ref: No injury)* | | | |
| Injury | -0.606* | 0.345 | 0.152 |
| | (0.258) | (0.248) | (0.271) |
| *Relationship status (Ref: Married/cohabiting* | | | |
| Single/widowed | -0.327* | -0.047 | -0.048 |
| | (0.128) | (0.246) | (0.252) |
| Separated/divorced | -0.620** | -1.494*** | -1.661* |
| | (0.196) | (0.450) | (0.696) |
| *Ethnicity (Ref: White)* | | | |
| Not White | -0.331 | 0.247 | -0.029 |
| | (0.199) | (0.346) | (0.426) |
| *Employment status (Ref: Employed)* | | | |
| Unemployed | 0.117 | 0.540 | 0.526 |
| | (0.120) | (0.345) | (0.345) |
| Outside labour force | -0.037 | -0.214 | -0.253 |
| | (0.167) | (0.269) | (0.288) |
| Student | -0.410* | -0.098 | -0.108 |
| | (0.191) | (0.452) | (0.374) |
| *Housing tenure (Ref: Homeowner)* | | | |
| Renter | 0.144 | 0.424 | 0.511 |
| | (0.137) | (0.238) | (0.320) |
| Other | 0.005 | 0.786 | 1.123* |
| | (0.120) | (0.418) | (0.523) |
| *Nr of dependent* | -0.323*** | -0.786*** | -0.796*** |
| | (0.056) | (0.197) | (0.218) |
| *Age* | 0.023*** | 0.024** | 0.029*** |
| | (0.004) | (0.009) | (0.007) |
| Constant | -2.424*** | -3.600*** | -3.917*** |
| | (0.232) | (0.589) | (0.430) |
| Observations (N) | 6,102 | 1,232 | 6,102 |

Source: based on CSEW and RCEW datasets

*** p<0.001

** p<0.01

* p<0.05. N = sample size.

Results presented as regression coefficients (for consistency) followed by standard errors (SE) in brackets.

to what we saw in our analyses of age, the imputed dataset mimics the associations from the CSEW dataset. For example, men were less likely to experience rape than women, while stranger perpetrators where more strongly associated with male than female victim-survivors. Two important things stand out: Acquaintance (compared to domestic relationship) perpetrator was not associated with gender, nor was 'other' housing tenure (compared to homeowners) in the CSEW, but these do become significant in the imputed dataset. The latter is most likely due to the far higher prevalence of 'other' housing tenure in the RCEW dataset, making it more likely to reach statistical significance, while the former is likely due to the larger sample size of the imputed dataset compared to the original CSEW dataset.

We considered the consistencies across Tables 2 and 3 as an indication that our proposed approach works for variables that are recorded similarly in the two datasets. We then extended our testing to an outcome measure which is *not* similarly recorded in CSEW and RCEW; health impact. For this, we specified the comparison vector as:

$$C_{i,j}^3 = f[\text{type of SV, perpetrator relationship, employment status,}$$

$$\text{housing tenure, number of dependants, relationship status, ethnicity, age, gender}] \quad (5)$$

In this scenario, a possible research question would be: what is the relationship between health impact (as a dependent variable) and type of sexual violence experienced, relationship to the perpetrator, health impact, employment status, housing tenure, number of dependants, relationship status, ethnicity, age and gender in the CSEW, in the RCEW and in the combined synthetic CSEW-RCEW datasets? Also in this scenario, we may be interested in examining the associations between (amongst others) the health impact and service needs., but health impact is not available in the target dataset; which is why we impute it here based on the CSEW. Table 4 shows the results of logistic regressions looking at the associations between health impact for dataset A (RCEW), dataset B (CSEW) and the complete combined synthetic dataset. We acknowledge that this is a more meaningful regression specification than the previous two specified in the paper. However, since our approach is novel, we wanted to ensure that the approach worked for similarly recorded variables before testing for differently recorded ones. The results show the same as for the previous models, namely that the imputed dataset reflects the associations from the CSEW dataset in both magnitude, direction, and significance. This includes an association that was positive in the original RCEW dataset (dataset A), but negative in CSEW (dataset B), and thus are also negative in the imputed dataset, namely, whether the perpetrator was a stranger or unknown. In these models, the coefficients for single/widowed stand out, given the synthetic dataset presents a very similar association to that found in the RCEW dataset. This is again likely a result of a much higher prevalence of this group in the RCEW (and therefore in the synthetic dataset).

Finally, in order to achieve our goal of combining data in a real-life application and producing a complete integrated dataset, we inputted a variable that only appears in CSEW and is, therefore, completely missing in RCEW; frequency of abuse. In this case, the comparison vector is:

$$C_{i,j}^4 = f[\text{type SV, perpetrator relationship, health impact, employment status,}$$

$$\text{housing tenure, number of dependants, relationship status, ethnicity, age, gender}] \quad (6)$$

In this scenario, a possible research question would be: what is the relationship between the frequency of abuse (as a dependent variable) and type of sexual violence experienced, relationship to the perpetrator, health impact, employment status, housing tenure, number of

**Table 4. Associations between health impact and other variables in RCEW data, CSEW data, and the imputed synthetic dataset.** Logistic regression models.

| | Dataset A: RCEW original | Dataset B: CSEW | Synthetic: Dataset A imputed dependent based on dataset B |
|---|---|---|---|
| | B(SE) | B(SE) | B(SE) |
| *Sexual violence* (Ref: Other) | | | |
| Rape | 0.187 | 1.507*** | 1.541*** |
| | (0.127) | (0.148) | (0.138) |
| *Victim-perpetrator relationship*(Ref: domestic) | | | |
| Acquaintance | 0.224 | -0.890*** | -0.720*** |
| | (0.121) | (0.178) | (0.122) |
| Stranger or unknown | 0.401* | -1.137*** | -0.992*** |
| | (0.168) | (0.181) | (0.163) |
| *Gender (Ref: Female)* | | | |
| Male | -0.611* | 0.232 | -0.004 |
| | (0.258) | (0.234) | (0.322) |
| *Relationship status (Ref: Married/cohabiting* | | | |
| Single/widowed | -0.305* | 0.325 | 0.324* |
| | (0.151) | (0.185) | (0.149) |
| Separated/divorced | -0.280 | 0.183 | 0.125 |
| | (0.206) | (0.221) | (0.241) |
| *Ethnicity (Ref: White)* | | | |
| Not White | 0.043 | -0.241 | -0.351 |
| | (0.187) | (0.254) | (0.309) |
| *Employment status (Ref: Employed)* | | | |
| Unemployed | 0.342* | 0.246 | 0.288 |
| | (0.138) | (0.260) | (0.265) |
| Outside labour force | 0.386* | 0.300 | 0.241 |
| | (0.189) | (0.157) | (0.212) |
| Student | 0.306 | -1.009** | -0.954** |
| | (0.183) | (0.348) | (0.362) |
| *Housing tenure (Ref: Homeowner)* | | | |
| Renter | -0.068 | 0.503** | 0.614* |
| | (0.149) | (0.157) | (0.260) |
| Other | -0.524*** | 0.346 | 0.625 |
| | (0.137) | (0.329) | (0.421) |
| *Nr of dependent* | 0.002 | 0.193** | 0.178** |
| | (0.051) | (0.075) | (0.067) |
| *Age* | 0.006 | 0.005 | 0.010* |
| | (0.005) | (0.007) | (0.005) |
| Constant | -2.984*** | -1.133** | -1.415*** |
| | (0.281) | (0.353) | (0.219) |
| Observations (N) | 6,102 | 1,232 | 6,102 |

Source: based on CSEW and RCEW datasets

*** p<0.001

** p<0.01

* p<0.05 N = sample size.

Results presented as regression coefficients (for consistency) followed by standard errors (SE) in brackets.

dependants, relationship status, ethnicity, age and gender in the CSEW and in the combined synthetic CSEW-RCEW datasets? Also in this scenario, we may be interested in examining the associations between (amongst others) the frequency of the abuse and service needs., but frequency of the abuse is not available for RCEW which is why we impute it here based on the CSEW. The analyses, in this case, used negative binomial models, which were deemed most appropriate due to over dispersion of the count variable (frequency of sexual violence incidents or repetitions), its relative low incidence in the data and long tailed distribution, as well as minimal Akaike Information Criterion (AIC) and the Bayesian Information Criterion (BIC). The results of the negative binomial regressions (Table 5) estimating the number of sexual violence incidents or repetitions based on CSEW data reveals that rape (compared to other sexual violence) and incidents by acquaintances or strangers (compared to domestic perpetrators) are less likely to be repeated, and if repeated they are repeated fewer times. The imputed synthetic dataset reflects these associations. On the other hand, whilst in the CSEW, significant negative associations between sexual violence incidents and singles/widowed (versus married or cohabitors), non-White (compared to White) victim-survivors exist, these associations did not reach statistical significance in the imputed dataset. Lastly, while students did not have a higher number of sexual violence incidents compared to employed people in the CSEW, in the imputed synthetic dataset this was the case. This change in significance is likely due to the larger proportion of students in the RCEW (and therefore in the synthetic dataset).

## Discussion

First and foremost, the associations found between the imputed versions of age, gender and health impact and the other variables in the CSEW are similar to those explored in the literature [33–35]. For example, our analysis also found that women are more likely to experience sexual violence than men, and that if the violence is perpetrated by a domestic relation this is more likely to cause a health impact. Previous studies that have used the CSEW have also highlighted similar methodological/analytical/technical difficulties to those we encountered. Particularly, Skafida et al [36] point out that while sexual violence is robustly measured in the CSEW, incidents are infrequent and health impacts mostly focus on physical harm. Likewise, we were only able to look at physical health impact due to limited measurement of mental health impacts in the CSEW.

In terms of our regression results using RCEW data, we found fewer studies using the same dataset. This is mainly due to restricted data access as RCEW only shares their data with trusted research partners due to increased vulnerability of the victim-survivors they serve [37] (see Data Protection Impact Assessment for details in S1 Table). However, there were two studies that used Rape Crisis data quantitatively. Like the current study, Lovett and Kelly found women to be more likely to experience rape than men and ethnicity to be poorly recorded [30]. Again similar to the current study, Bunce et al compared those victim-survivors who engaged with the service with those who disengaged, and found instability/vulnerability with regards to housing tenure and employment status to be negatively associated with engagement. [38]. There are several implications from our proposed approach to combining data, based on look-alike principles, using multiple imputation methods. First, the initial distribution may be different between datasets, as was the case in the RCEW and CSEW, and while this does not appear to prevent meaningful analyses in the synthetic dataset, the sample sizes are important both in defining what dataset ultimately provides the basis for the synthetic dataset and also in interpreting some of the meaningful associations found. In general, in our proposed exercise, the associations mimic those of the CSEW (smaller sample size), which was used as the basis for imputation. However, where the prevalence of a certain group was much

**Table 5. Associations between number of incidents or repetitions and other variables in CSEW, and the imputed synthetic dataset.** Negative binomial models.

| | Dataset B: CSEW | Synthetic: Dataset A imputed dependent based on dataset B |
|---|---|---|
| | B(SE) | B(SE) |
| *Sexual violence* (Ref: Other) | | |
| Rape | -0.585* | -0.706* |
| | (0.262) | (0.329) |
| *Victim-perpetrator relationship*(Ref: domestic) | | |
| Acquaintance | -1.560*** | -1.567*** |
| | (0.275) | (0.324) |
| Stranger or unknown | -2.764*** | -2.742*** |
| | (0.299) | (0.398) |
| *Gender (Ref: Female)* | | |
| Male | -0.539 | -0.372 |
| | (0.400) | (0.558) |
| *Health impact (Ref: No injury)* | | |
| Injury | 0.313 | 0.402 |
| | (0.249) | (0.474) |
| *Relationship status (Ref: Married/cohabiting* | | |
| Single/widowed | -0.531* | -0.649 |
| | (0.267) | (0.377) |
| Separated/divorced | 0.046 | 0.086 |
| | (0.316) | (0.269) |
| *Ethnicity (Ref: White)* | | |
| Not White | -0.869* | -0.977 |
| | (0.389) | (0.543) |
| *Employment status (Ref: Employed)* | | |
| Unemployed | 0.129 | -0.030 |
| | (0.405) | (0.354) |
| Outside labour force | 0.148 | 0.166 |
| | (0.255) | (0.516) |
| Student | 0.593 | 0.796* |
| | (0.510) | (0.363) |
| *Housing tenure (Ref: Homeowner)* | | |
| Renter | 0.086 | 0.081 |
| | (0.233) | (0.484) |
| Other | 0.083 | 0.097 |
| | (0.566) | (0.624) |
| *Nr of dependent* | -0.041 | -0.006 |
| | (0.120) | (0.112) |
| *Age* | 0.017 | 0.010 |
| | (0.010) | (0.011) |
| lnalpha | 2.166*** | 2.168*** |
| | (0.084) | (0.138) |
| Constant | 1.135* | 1.339 |
| | (0.525) | (0.684) |

(*Continued*)

**Table 5.** (Continued)

| | Dataset B: CSEW | Synthetic: Dataset A imputed dependent based on dataset B |
|---|---|---|
| | B(SE) | B(SE) |
| Observations | 1,217 | 6,102 |

Source: based on CSEW and RCEW datasets

\*\*\* p<0.001

\*\* p<0.01

\* p<0.05

larger in the RCEW (larger sample size), this group was also larger in the synthetic version, meaning that there was an increased chance of significance. In order to test the robustness of our approach, we swapped datasets A and B, that is, we tested imputing data from the RCEW into the CSEW. This led to a synthetic dataset that was the size of the CSEW (1,232). While we found the same general findings, i.e. that magnitude and direction of effect sizes in the synthetic dataset mimicked those of the RCEW (used for imputation instead), standard errors were in general larger, meaning results were less likely to reach significance. This reinforces the importance of sample sizes (both in the imputing and in the imputed datasets).

A strength of the proposed method is that it enables the combining of data on different individuals based on similar characteristics, meaning that working with pseudonymised data is possible. This is relevant to any area of research where there are concerns around data-sharing, not only violence. Having said this, the novelty of our study lies in its application to violence research, by proposing the use of well-established methods in data science (i.e. creating a synthetic dataset using multiple imputation) and in combining the two datasets we used in this study. The combined RCEW-CSEW synthetic data would, for instance, enable novel multisectorial analysis, including potentially mental health impacts (well recorded in RCEW but not in CSEW) at population level, which have been scarce due to limitations of the CSEW [36] or the experience of (sexual violence) threats (well recorded in CSEW but not in RCEW) at practice level, which to date have been hindered by data access in this field [37]. Furthermore, our analyses have shown that results are fairly consistent regardless of the type of modelling used (OLS, logistic or negative binomials). Integrated survey and administrative data can strengthen study designs by providing more complete information on similar profiles, lessening response burden on participants, or by serving as a source of triangulated data [39].

The approach outlined involved a trade-off between the standardisation of variables required for imputation and the detail about individuals and experiences that is valued in research on violence. The need to standardise variables used for imputation meant that more nuanced understanding of experiences was lost. In our analyses, this was particularly relevant in terms of health impact. While our final coding only allowed for the inclusion of a binary, there is a wide literature on the impacts of sexual violence on physical health [40–42], and some of the final categories in the variables we used were much more aggregate than we would have liked. This was also the case for ethnicity, precluding analyses using an intersectional approach. Furthermore, we did not consider *time* (i.e. time of experience of sexual violence) as a variable in the comparison vector due to limited sample sizes, but we acknowledge that the understanding of experiences of violence varies over time, so ideally *time* should be a comparison-vector variable.

Our proposed data integration approach should be particularly useful for costing or burden of disease type of analyses, including calculating the societal burden of violence, given it enables taking a micro-costing approach, which produces more precise estimates [43].

Nonetheless, further applications, in particular to evaluate interventions, need further testing. Analyses using a longitudinal design are certainly not feasible if *time* is not used as a comparison-vector variable.

Similarly to all applications of multiple imputation, there are assumptions around the patterns of data missingness. While MICE assumes data missing at random (MAR) or missing completely at random (MCAR), when using our approach to impute a variable that only appears in one dataset, there is a normative assumption that the synthetic dataset follows the same distribution (and the same pattern of missingness) as the dataset used for imputation. Furthermore, while we chose MICE [44] as a method for imputation due to its easy implementation and widespread use in data completely missing [45, 46], we could have used other methods for imputation, including deep neural network methods [47] and Gaussian processes for non-parametric models [48]. While both deep neural networks and Gaussian processes are more flexible than MICE, they usually require larger datasets for deep learning [49].

Internationally, literature on recent approaches to data integration have gained relevance and have been covered in a Special Edition by the Journal of Survey Statistics and Methodologies [50], including ethical issues around direct and probabilistic data linkage and other methods for data integration. In total, the special edition published twelve papers on this topic, with four different applications combining survey and administrative data. More comparable to our study is the method proposed by Moretti and Shlomo, which combined information on multiple social domains, such as social exclusion and wellbeing, and provided applications using the European Union Statistics for Income and Living Conditions and Living Costs and Food Survey for the United Kingdom [51]. Like us, the authors see the application of integration methods to social sciences (including violence) as a future opportunity for research.

Finally, there are numerous practice and policy implications for researchers, voluntary sector partner organisations, and the general population. Compared to traditional research, our proposed approach to data integration offers a cost-effective solution to breaking (data-related) silos in research. Further research should not only test different approaches to data integration, but also applications to evaluations by mutually engaging practitioners, policy-makers, and researchers to foster a culture of research [39, 52] facilitating the refinement of techniques as well as producing real-world evidence based on integrated synthetic data.

## Conclusion

This study has demonstrated that data integration between a survey (CSEW) and administrative records (RCEW) is possible using look-alike modelling principles and using multiple imputation by chained equations. Our results serve as a proof of concept, and the associations in the resulting synthetic dataset tend to mimic the dataset used for imputation in magnitude and direction. The regression results in the synthetic dataset also tend to yield larger standard errors, resulting in larger confidence intervals. This approach should be applicable for costing exercises as it permits micro-costing. Further applications of the approach should be the focus of future research.

## Supporting information

**S1 Table. Variable harmonising across the CSEW and RCEW.**
(DOCX)

## Author Contributions

**Conceptualization:** Estela Capelas Barbosa.

**Data curation:** Niels Blom, Annie Bunce.

**Formal analysis:** Estela Capelas Barbosa, Niels Blom.

**Funding acquisition:** Estela Capelas Barbosa.

**Methodology:** Estela Capelas Barbosa.

**Supervision:** Estela Capelas Barbosa.

**Validation:** Niels Blom.

**Visualization:** Estela Capelas Barbosa, Niels Blom, Annie Bunce.

**Writing – original draft:** Estela Capelas Barbosa.

**Writing – review & editing:** Estela Capelas Barbosa, Niels Blom, Annie Bunce.

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
