## [Decision Letter · Decision Letter 0]

16 Jul 2024

PONE-D-24-07569Look-alike modelling in violence-related research: a missing data approachPLOS ONE

Dear Dr. Barbosa,

Thank you for submitting your manuscript to PLOS ONE. After careful consideration, we feel that it has merit but does not fully meet PLOS ONE’s publication criteria as it currently stands. Therefore, we invite you to submit a revised version of the manuscript that addresses the points raised during the review process.

**ACADEMIC EDITOR** Let the abstract flow like this: background, aim, method, result, and conclusion.Confirm that the data and proposed models are sufficient for this research. Are there any limitations for the model or the nature of the data that could have been a concern?Are there any constraints that were relaxed for the research?Why was negative binomial used instead of Poisson?Compare this result with others that used the same dataset and discuss accordingly.Was the adopted missing data analysis adequate for the research?Discuss the result with those in similar countries: the US, Canada, and the EU.==============================

We look forward to receiving your revised manuscript.

Kind regards,

Academic Editor

PLOS ONE

Journal Requirements:

"This paper is a result of VISION research, which is supported by the UK Prevention Research Partnership (Violence, Health and Society; MR-VO49879/1). VISION is a Consortium funded by the British Heart Foundation, Chief Scientist Office of the Scottish Government Health and Social Care Directorates, Engineering and Physical Sciences Research Council, Economic and Social Research Council, Health and Social Care Research and Development Division (Welsh Government), Medical Research Council, National Institute for Health and Care Research, Natural Environment Research Council, Public Health Agency (Northern Ireland), The Health Foundation, and Wellcome. The views expressed are those of the researchers and not necessarily those of the UK Prevention Research Partnership or any other funder."

3. In the online submission form, you indicated that "The data are not publicly available due to restrictions agreed in the data sharing process with Rape Crisis England and Wales, due to concerns for the safety of their service users. The data that support the findings of this study can be made available on reasonable request from the corresponding author, ECB, if consented by Rape Crisis England and Wales."

5. We note you have included a table to which you do not refer in the text of your manuscript. Please ensure that you refer to Table 5 in your text; if accepted, production will need this reference to link the reader to the Table.

Reviewers' comments:

Reviewer's Responses to Questions

**Comments to the Author**

1. Is the manuscript technically sound, and do the data support the conclusions?

Reviewer #1: Yes

Reviewer #2: Yes

Reviewer #3: Yes

2. Has the statistical analysis been performed appropriately and rigorously? 

Reviewer #1: N/A

Reviewer #2: I Don't Know

Reviewer #3: Yes

3. Have the authors made all data underlying the findings in their manuscript fully available?

Reviewer #1: No

Reviewer #2: No

Reviewer #3: Yes

4. Is the manuscript presented in an intelligible fashion and written in standard English?

Reviewer #1: Yes

Reviewer #2: Yes

Reviewer #3: Yes

5. Review Comments to the Author

Reviewer #1: The approach described in the current study, which involves treating data integration and look-alike profiling as a missing data problem and using multiple imputation with chained equations to create a synthetic dataset, is not entirely new. However, its application in specific contexts, like combining survey data with administrative data from Rape Crisis Centres to focus on victim-survivors of sexual violence, could be considered innovative and valuable in certain fields.

Treating data integration as a missing data problem is a recognized method in statistical and data science literature. Multiple imputation with chained equations (MICE) is a well-established technique for handling missing data.

What might be novel is the specific application of this technique to the integration of the Crime Survey for England and Wales with administrative data from Rape Crisis Centres.

The creation of a synthetic dataset that integrates survey and administrative data specifically for understanding and supporting adult victim-survivors of sexual violence might provide new opportunities for research and policy-making.

While the methods used (data integration as a missing data problem and multiple imputation with chained equations) are established in the literature, their application to the specific context of combining survey and administrative data on victim-survivors of sexual violence might offer new insights and could be considered innovative in this field. The novelty lies more in the application and the potential impact on research and policy rather than in the methods themselves.

Line 78-80: Literature study on missing value imputation is not sufficient. Missing reference to MICE algorithm? Why MICE while there are many other methods for missing value imputation?

For example:

Lin, W. C., Tsai, C. F., & Zhong, J. R. (2022). Deep learning for missing value imputation of continuous data and the effect of data discretization. Knowledge-Based Systems, 239, 108079.

Jafrasteh, B., Hernández-Lobato, D., Lubián-López, S. P., & Benavente-Fernández, I. (2023). Gaussian processes for missing value imputation. Knowledge-Based Systems, 273, 110603.

Reviewer #2: Thank you. This is clearly in depth work. I think efforts to make the method more intelligible to readers would be beneficial. Please could you include an intuitive explanation of what you did and why in the abstract?

My understanding is that the idea was to implement MI one dataset by borrowing distributional knowledge from another; could you clarify for readers why this might be useful, e.g. including examples? I think in this regard the concept of data integration needs more careful definition for non statistical/data science readers. Regarding the specification of the vector- isn't the configuration of an appropriate imputation model dependent on the question, and should a specific research question be included? Sample sizes in all tables(and other information so that they can be read in isolation) would help a lot I think. I think from a less statistical and more epidemiological standpoint, this might be more useful to specific a primary analysis of a research question in one dataset, and implement the approach described to generate another analytic dataset on the same variables, so readers can consider whether the approach delivers benefit. As it is the benefits for future work is not clear.

Reviewer #3: The study is unique, as it used one of the scientific models that will contribute to the development of scientific research in social fields through the use of integrated data. The researchers applied appropriate statistics to the measurement levels of the dependent variables in the study.

6. PLOS authors have the option to publish the peer review history of their article (what does this mean?). If published, this will include your full peer review and any attached files.

Reviewer #1: No

Reviewer #2: No

Reviewer #3: **Yes: **Professor Hussain Al-Othman, University of Sharjah, U.A.E

---

## [Author Response · Author response to Decision Letter 0]

10 Dec 2024

Responses to Academic Editor and Reviewers

Comments from Academic Editor

Let the abstract flow like this: background, aim, method, result, and conclusion- We have amended the abstract to flow as suggested. See page 2 (lines 20-47).

Confirm that the data and proposed models are sufficient for this research. Are there any limitations for the model or the nature of the data that could have been a concern? - Data and proposed models are sufficient for this research. We have added a sentence to this effect. New text (lines 191-193) “These two datasets (CSEW and RCEW) are sufficient to achieve our aim of creating a combined synthetic dataset and no statistical constraints were relaxed while conducting our empirical application.”

Are there any constraints that were relaxed for the research? - No constraints were relaxed for this research. We have added a sentence to this effect. New text (lines 200-202) “These two datasets (CSEW and RCEW) are sufficient to achieve our aim of creating a combined synthetic dataset and no statistical constraints were relaxed while conducting our empirical application.”

Why was negative binomial used instead of Poisson? - Data in CSEW and RCEW are over dispersed, thus, Poison regressions would be inappropriate for count variables. We have tested fit considering AIC and BIC and both were minimal for negative binomials. New text (lines 381-384) “The analyses, in this case, used negative binomial models, which were deemed most appropriate due to over dispersion of the count variable (frequency of sexual violence incidents or repetitions), its relative low incidence in the data and long tailed distribution, as well as minimal Akaike Information Criterion (AIC) and the Bayesian Information Criterion (BIC).”

Compare this result with others that used the same dataset and discuss accordingly. - We have added a paragraph to discuss the results from the same datasets, even though the aim of our paper was not exactly to produce new evidence using either. We highlighted some of the literature and discussed accordingly. New text (lines 398-417) “First and foremost, the associations found between the imputed versions of age, gender and health impact and the other variables in the CSEW are similar to those explored in the literature [33-35] . For example, our analysis also found that women are more likely to experience sexual violence than men, and that if the violence is perpetrated by a domestic relation this is more likely to cause a health impact. Previous studies that have used the CSEW have also highlighted similar methodological/analytical/technical difficulties to those we encountered. Particularly, Skafida et al [36] point out that while sexual violence is robustly measured in the CSEW, incidents are infrequent and health impacts mostly focus on physical harm. Likewise, we were only able to look at physical health impact due to limited measurement of mental health impacts in the CSEW. In terms of our regression results using RCEW data, we found fewer studies using the same dataset. This is mainly due to restricted data access as RCEW only shares their data with trusted research partners due to increased vulnerability of the victim-survivors they serve [37] (see Data Protection Impact Assessment for details in Supplementary files). However, there were two studies that used Rape Crisis data quantitatively. Like the current study, Lovett and Kelly found women to be more likely to experience rape than men and ethnicity to be poorly recorded [30]. Again similar to the current study, Bunce et al compared those victim-survivors who engaged with the service with those who disengaged, and found instability/vulnerability with regards to housing tenure and employment status to be negatively associated with engagement. [38].”

Was the adopted missing data analysis adequate for the research? - We used MICE due to its easy implementation and wide use in the literature for dealing with data completely missing, but we acknowledge that other techniques for dealing with missing data could have been used. We included a sentence to clarify that other approaches could have also been adopted. New text (lines 472-477) “Furthermore, while we chose MICE [44] as a method for imputation due to its easy implementation and widespread use in data completely missing [45, 46], we could have used other methods for imputation, including deep neural network methods [47] and Gaussian processes for non-parametric models [48]. While both deep neural networks and Gaussian processes are more flexible than MICE, they usually require larger datasets for deep learning [49].”

Discuss the result with those in similar countries: the US, Canada, and the EU. - We have added a paragraph, discussing international literature on the issue of data integration. New text (lines 478-488) “Internationally, literature on recent approaches to data integration have gained relevance and have been covered in a Special Edition by the Journal of Survey Statistics and Methodologies [50], including ethical issues around direct and probabilistic data linkage and other methods for data integration. In total, the special edition published twelve papers on this topic, with four different applications combining survey and administrative data. More comparable to our study is the method proposed by Moretti and Shlomo, which combined information on multiple social domains, such as social exclusion and wellbeing, and provided applications using the European Union Statistics for Income and Living Conditions and Living Costs and Food Survey for the United Kingdom [51]. Like us, the authors see the application of integration methods to social sciences (including violence) as a future opportunity for research.”

Comments from Reviewers 1 

The approach described in the current study, which involves treating data integration and look-alike profiling as a missing data problem and using multiple imputation with chained equations to create a synthetic dataset, is not entirely new. However, its application in specific contexts, like combining survey data with administrative data from Rape Crisis Centres to focus on victim-survivors of sexual violence, could be considered innovative and valuable in certain fields. - Indeed, the novelty is the application and in the combination of these two specific datasets. We have added a sentence to highlight this. New text (lines 437-440). “Having said this, the novelty of our study lies in its application to violence research, by proposing the use of well-established methods in data science (i.e. creating a synthetic dataset using multiple imputation) and in combining the two datasets we used in this study”

Treating data integration as a missing data problem is a recognized method in statistical and data science literature. Multiple imputation with chained equations (MICE) is a well-established technique for handling missing data. What might be novel is the specific application of this technique to the integration of the Crime Survey for England and Wales with administrative data from Rape Crisis Centres. - Once again, we added a sentence for clarity as it is indeed the integration of CSEW and RCEW that may be considered novel. New text (lines 440-444) “The combined CSEW-RCEW synthetic data would, for instance, enable novel multi-sectorial analysis, including potentially mental health impacts (well recorded in RCEW but not in CSEW) at population level, which have been scarce due to limitations of the CSEW [36] or the experience of (sexual violence) threats (well recorded in CSEW but not in RCEW) at practice level, which to date have been hindered by data access in this field [37].”

Line 78-80: Literature study on missing value imputation is not sufficient. Missing reference to MICE algorithm? Why MICE while there are many other methods for missing value imputation?

For example:

Lin, W. C., Tsai, C. F., & Zhong, J. R. (2022). Deep learning for missing value imputation of continuous data and the effect of data discretization. Knowledge-Based Systems, 239, 108079.

Jafrasteh, B., Hernández-Lobato, D., Lubián-López, S. P., & Benavente-Fernández, I. (2023). Gaussian processes for missing value imputation. Knowledge-Based Systems, 273, 110603. We agree we could have used other imputation methods, so we added not only reference to the MICE algorithm but also justification for our decision to implement it. 

The revised manuscript includes this justification. New text (lines 472-477) “Furthermore, while we chose MICE [44] as a method for imputation due to its easy implementation and widespread use in data completely missing [45, 46], we could have used other methods for imputation, including deep neural network methods [47] and Gaussian processes for non-parametric models [48]. While both deep neural networks and Gaussian processes are more flexible than MICE, they usually require larger datasets for deep learning [49].”

Comments from Reviewers 2 

Thank you. This is clearly in depth work. I think efforts to make the method more intelligible to readers would be beneficial. Please could you include an intuitive explanation of what you did and why in the abstract? - 

We have added a sentence with an intuitive explanation both to the abstract and to the methods section. New text (lines 29-32 and 147-150) “Intuitively, the idea was to impute missing information from one dataset by borrowing the distribution from the other. In our analyses, we borrowed information from CSEW to impute missing data in the RCEW administrative dataset, creating a combined synthetic RCEW-CSEW dataset.”

My understanding is that the idea was to implement MI one dataset by borrowing distributional knowledge from another; could you clarify for readers why this might be useful, e.g. including examples? - Your understanding is correct. The idea of borrowing the distribution from another dataset enables new analyses that to date have been hindered by data access. We have added a paragraph with a couple of examples. New text (lines 437-444) “Having said this, the novelty of our study lies in its application to violence research, by proposing the use of well-established methods in data science (i.e. creating a synthetic dataset using multiple imputation) and in combining the two datasets we used in this study. The combined RCEW-CSEW synthetic data would, for instance, enable novel multi-sectorial analysis, including potentially mental health impacts (well recorded in RCEW but not in CSEW) at population level, which have been scarce due to limitations of the CSEW [36] or the experience of (sexual violence) threats (well recorded in CSEW but not in RCEW) at practice level, which to date have been hindered by data access in this field [37].”

I think in this regard the concept of data integration needs more careful definition for non statistical/data science readers. Regarding the specification of the vector- isn't the configuration of an appropriate imputation model dependent on the question, and should a specific research question be included? 

We have added specific possible research questions to all our four empirical applications. New text for application 1 (lines 292-298): “In this scenario, a possible research question would be: what is the relationship between age (as a dependent variable) and type of sexual violence experienced, relationship to the perpetrator, health impact, employment status, housing tenure, number of dependants, relationship status, ethnicity and gender in the RCEW, in the CSEW and in the combined synthetic RCEW-CSEW datasets? More realistically, such an imputed dataset could be used to answer questions such as how is age related to type of sexual violence victimisation among people accessing specialist support services. For application 2 (lines 322-328) “In this scenario, a possible research question would be: what is the relationship between gender (as a dependent variable) and type of sexual violence experienced, relationship to the perpetrator, health impact, employment status, housing tenure, number of dependants, relationship status, ethnicity and age in the RCEW, in the CSEW and in the combined synthetic RCEW-CSEW datasets? More realistically, such an imputed dataset could be used to answer questions such as how is gender related to type of sexual violence victimisation among people accessing specialist support services.” For application 3 (lines 347-353): “In this scenario, a possible research question would be: what is the relationship between health impact (as a dependent variable) and type of sexual violence experienced, relationship to the perpetrator, health impact, employment status, housing tenure, number of dependants, relationship status, ethnicity, age and gender in the CSEW, in the RCEW and in the combined synthetic CSEW-RCEW datasets? Also in this scenario, we may be interested in examining the associations between (amongst others) the health impact and service needs., but health impact is not available in the target dataset; which is why we impute it here based on the CSEW.” For application 4 (lines 374-385): “In this scenario, a possible research question would be: what is the relationship between the frequency of abuse (as a dependent variable) and type of sexual violence experienced, relationship to the perpetrator, health impact, employment status, housing tenure, number of dependants, relationship status, ethnicity, age and gender in the CSEW and in the combined synthetic CSEW-RCEW datasets? Also in this scenario, we may be interested in examining the associations between (amongst others) the frequency of the abuse and service needs., but frequency of the abuse is not available for RCEW which is why we impute it here based on the CSEW.”

Sample sizes in all tables(and other information so that they can be read in isolation) would help a lot I think.- We have included information on sample sizes. We have also added a note below each table to explain that results are presented by regression coefficients (for consistency) and standard errors (SE) in brackets. See tables.

I think from a less statistical and more epidemiological standpoint, this might be more useful to specific a primary analysis of a research question in one dataset, and implement the approach described to generate another analytic dataset on the same variables, so readers can consider whether the approach delivers benefit. As it is the benefits for future work is not clear.- Indeed the paper is proposing a methodology for data integration and our examples may not be particularly meaningful for practice, although they were designed as proof of concept. We have added two practical examples where our approach may be beneficial for applications. New text (lines 440-444) “The combined CSEW-RCEW synthetic data would, for instance, enable novel multi-sectorial analysis, including potentially mental health impacts (well recorded in RCEW but not in CSEW) at population level, which have been scarce due to limitations of the CSEW [36] or the experience of (sexual violence) threats (well recorded in CSEW but not in RCEW) at practice level, which to date have been hindered by data access in this field [37].”

Comments from Reviewers 3 

The study is unique, as it used one of the scientific models that will contribute to the development of scientific research in social fields through the use of integrated data. The researchers applied appropriate statistics to the measurement levels of the dependent variables in the study. - Thank you for your consideration of our study and for your positive review.

---

## [Decision Letter · Decision Letter 1]

23 Dec 2024

Look-alike modelling in violence-related research: a missing data approach

PONE-D-24-07569R1

Dear Dr. Barbosa,

We’re pleased to inform you that your manuscript has been judged scientifically suitable for publication and will be formally accepted for publication once it meets all outstanding technical requirements.

Kind regards,

Hilary Izuchukwu Okagbue, Ph.D

Academic Editor

PLOS ONE

Additional Editor Comments (optional):

Reviewers' comments:

Reviewer's Responses to Questions

**Comments to the Author**

1. If the authors have adequately addressed your comments raised in a previous round of review and you feel that this manuscript is now acceptable for publication, you may indicate that here to bypass the “Comments to the Author” section, enter your conflict of interest statement in the “Confidential to Editor” section, and submit your "Accept" recommendation.

Reviewer #1: All comments have been addressed

2. Is the manuscript technically sound, and do the data support the conclusions?

Reviewer #1: Yes

3. Has the statistical analysis been performed appropriately and rigorously? 

Reviewer #1: N/A

4. Have the authors made all data underlying the findings in their manuscript fully available?

Reviewer #1: No

5. Is the manuscript presented in an intelligible fashion and written in standard English?

Reviewer #1: Yes

6. Review Comments to the Author

Reviewer #1: Thanks. The author addressed my comments and revised the paper.

It can be published in the current form.

7. PLOS authors have the option to publish the peer review history of their article (what does this mean?). If published, this will include your full peer review and any attached files.

Reviewer #1: No

---

## [Editor Report · Acceptance letter]

3 Jan 2025

PONE-D-24-07569R1 

PLOS ONE

Dear Dr. Barbosa, 

I'm pleased to inform you that your manuscript has been deemed suitable for publication in PLOS ONE. Congratulations! Your manuscript is now being handed over to our production team.

Kind regards, 

on behalf of

Dr Hilary Izuchukwu Okagbue 

Academic Editor

PLOS ONE